# Air-GR: An Over-the-Air Handwritten Character Recognition System Based on Coordinate Correction YOLOv5 Algorithm and LGR-CNN

**DOI:** 10.3390/s23031464

**Published:** 2023-01-28

**Authors:** Yajun Zhang, Zijian Li, Zhixiong Yang, Bo Yuan, Xu Liu

**Affiliations:** School of Software Engineering, Xinjiang University, Ürümqi 830046, China

**Keywords:** Air-GR, YOLOv5, LGR-CNN, gesture coordinate correction algorithm, time window algorithm, gesture recognition

## Abstract

Traditional human-computer interaction technology relies heavily on input devices such as mice and keyboards, which limit the speed and naturalness of interaction and can no longer meet the more advanced interaction needs of users. With the development of computer vision (CV) technology, research on contactless gesture recognition has become a new research hotspot. However, current CV-based gesture recognition technology has the limitation of a limited number of gesture recognition and cannot achieve fast and accurate text input operations. To solve this problem, this paper proposes an over-the-air handwritten character recognition system based on the coordinate correction YOLOv5 algorithm and a lightweight convolutional neural network (LGR-CNN), referred to as Air-GR. Unlike the direct recognition of captured gesture pictures, the system uses the trajectory points of gesture actions to generate images for gesture recognition. Firstly, by combining YOLOv5 with the gesture coordinate correction algorithm proposed in this paper, the system can effectively improve gesture detection accuracy. Secondly, considering that the captured gesture coordinates may contain multiple gestures, this paper proposes a time-window-based algorithm for segmenting the gesture coordinates. Finally, the system recognizes user gestures by plotting the segmented gesture coordinates in a two-dimensional coordinate system and feeding them into the constructed lightweight convolutional neural network, LGR-CNN. For the gesture trajectory image classification task, the accuracy of LGR-CNN is 13.2%, 12.2%, and 4.5% higher than that of the mainstream networks VGG16, ResNet, and GoogLeNet, respectively. The experimental results show that Air-GR can quickly and effectively recognize any combination of 26 English letters and numbers, and its recognition accuracy reaches 95.24%.

## 1. Introduction

In recent years, with the development of human-computer interaction (HCI) technology, the interaction between computers and people has become more and more frequent. The traditional HCI method mainly uses wearable devices such as mice, keyboards, touch screens, and data gloves, but this contact method has certain defects and requires additional hardware devices, which cannot meet the natural interaction needs. At the same time, the traditional human-computer interaction method requires contact with the device during use, which poses health and safety hazards.

With the gradual maturation of speech recognition [1], gesture recognition [2], and other related technologies, contactless interaction means have gradually become a popular direction of research. Hand gestures have gradually attracted the attention of researchers because of their intuitive image, easy understanding, and simple and flexible application. In addition, gestures can express various information, so there is significant potential in real-life applications, such as in medical treatment [3], where doctors can perform surgical treatment through gesture sensing; in smart homes [4], where people can control household devices through gesture movements; and in the VR field, where people can control games through gestures [5] to enhance the immersion of games. Therefore, it is important to study gesture recognition.

The current mainstream gesture recognition usually uses wearable sensors, radar, Wi-Fi, RFID, and other technologies. Although these methods can achieve good recognition rates in a fixed environment, they also have obvious drawbacks and limitations. For example, gesture recognition techniques based on wearable sensors [6] require users to wear corresponding devices while drawing gestures, and users inevitably forget or are inconvenienced wearing the devices. Radar-based gesture recognition techniques [7] have high sensitivity but are difficult to use commercially due to the high price of their devices. Although RFID [8] and Wi-Fi devices [9] are inexpensive, have wide signal coverage, and play a good role in gesture recognition, their accuracy rate is low in more complex multi-path environments.

In recent years, CV-based gesture recognition technology has attracted a lot of research interest due to its passive and non-contact characteristics. This means that the user does not need to carry any device for gesture detection and there is no problem with multi-path environmental interference. Although CV technology has shown considerable advantages in gesture recognition, there are still limitations. First, most systems mainly capture each frame of the user’s hand gesture movement process through a camera, after which the gestures in the pictures are extracted by algorithms such as edge detection or bone localization and sent into a trained neural network for recognition. This method requires evaluation of the 3D features of the gesture, including the shape, angle, and size of the hand during the gesture movement. To achieve a high accuracy rate, it is necessary to increase the depth of the model and train it with a large amount of data to support the expected results, which is not conducive to real-time gesture recognition. Secondly, existing studies are conducted in ideal environments, i.e., the captured gesture pictures contain only user gestures and do not consider the impact of other people’s gestures on the system. Finally, the existing systems also have the problem of a limited number of gesture recognition, which cannot meet the needs of practical applications. To solve the above problems, this paper proposes a CV-based over-the-air handwritten character recognition system Air-GR. The model consists of three parts: the gesture trajectory capture module, continuous gesture segmentation, the processing module, and the gesture drawing and recognition module. The main idea is to capture the user’s gesture coordinates through the target detection algorithm, segment and process the captured coordinates, then draw them in a two-dimensional coordinate system and input them into the constructed LGR-CNN network to complete the recognition. There are four challenges in designing the Air-GR system.

Challenge 1: How to capture the user’s gesture coordinates quickly and accurately?

Challenge 2: When users are drawing gestures, if there are other people walking around, the camera may detect other people’s gestures, thus causing interference to the collected gesture coordinates, so how to remove the interference from the surrounding people?

Challenge 3: Since the gesture coordinates captured by the model may contain multiple gestures, how to accurately segment a single gesture from the continuous gestures?

Challenge 4: How to perform fast and accurate recognition of the segmented gesture coordinates?

To resolve the above challenges, the proposed solution is described in Section 3, along with the feasibility of the method demonstrated through preliminary experiments. Next, the algorithm presented in Section 4 was used to assess the performance of the system through extensive experiments. The main contributions of this paper are listed as follows:(1)In this paper, we propose an over-the-air handwritten character recognition system Air-GR based on coordinate correction YOLOv5 algorithm and lightweight convolutional neural network LGR-CNN, which can effectively recognize any combination of English letters and numbers with lengths of less than 7. Unlike the direct recognition of captured gesture images, we use the trajectory points of gesture actions to generate images for gesture recognition, which can effectively reduce the model complexity and improve the speed and accuracy of gesture recognition.(2)To remove the gesture interference from surrounding people, this paper proposes a gesture coordinate correction algorithm, which can effectively improve the gesture detection accuracy by combining the algorithm with the YOLOv5 algorithm.(3)To realize the segmentation of continuous gestures, this paper proposes a time window-based continuous gesture segmentation algorithm, which obtains the start and end of a single gesture by computing the set of coordinate points with a slope close to 0 within the time window.(4)To improve the gesture recognition accuracy, this paper improves the structure of the traditional convolutional neural network and builds a lightweight convolutional neural network LGR-CNN for the gesture trajectory picture classification task, which was 13.2%, 12.2%, and 4.5% more accurate than VGG16, ResNet, and GoogLeNet mainstream networks, respectively.

## 2. Related Work

The existing gesture recognition technologies mainly include three categories: wearable sensors, wireless signals, and CV.

### 2.1. Wearable Sensor-Based Gesture Recognition Techniques

Wearable sensor-based approaches mainly use sensing devices to capture user hand movements and finger movements. Xianzhi Chu et al. [10] proposed an algorithm for Japanese Sign Language (JSL) recognition, which used a sensing glove to complete data acquisition and testing, and tested seven gestures with good recognition results. Xiaoliang Zhang et al. [11] who detected gestures by combining a wearable armband and a smart glove made from a customizable array of pressure sensors, achieved the recognition of 10 gestures. Haneul Jeon et al. [12] proposed a novel gesture recognition method based on wearable IMU sensors and experimentally demonstrated that the method is appropriate for gesture recognition under significant changes in the subject’s body alignment during gestures. In addition, to improve the accuracy of wearable sensor-based gesture recognition algorithms, Haneul Jeon et al. [13] proposed a DCGAN structure with a mode switcher for data enhancement of time-series sensor data. Although the sensor-based approach can sense gesture activity with high accuracy, the approach is not convenient because it requires the user to wear the corresponding device while drawing the gesture, and the user inevitably forgets or is not comfortable wearing the device. In addition, the sensor-based method suffers from high hardware costs and limited battery life of the sensor.

### 2.2. Gesture Recognition Techniques Based on Wireless Signal

The current gesture recognition technology based on wireless signals mainly captures gesture signals through WiFi, RFID, radar, and other devices to complete gesture recognition. Tang et al. [14] designed a WiFi gesture recognition system based on a parallel LSTM-FCN [15] neural network, extracted different dimensions of gesture by parallel long short-term memory full convolutional network (LSTM-FCN) model features, and evaluate 50 gestures with good accuracy. Tianzhang Xing et al. [16] designed a lightweight gesture recognition system, WiFine, based on Wi-Fi to achieve fast recognition of various actions. Yongpan Zou et al. [17] proposed GRfid, a novel device-free gesture recognition system based on phase information output by COTS RFID devices, to obtain gesture information in a non-contact, non-infringing manner by using RFID tags and achieve better accuracy on the test set. Wei Li et al. [18] proposed a millimeter-wave radar-based method for aerial handwritten digit recognition, which uses the trajectory points of gesture actions to generate images for gesture recognition, which is similar to the model proposed in this paper, however, the method can only recognize digits and is difficult to use commercially due to the high price of radar. Although WiFi and RFID devices are inexpensive and achieve acceptable recognition accuracy, the signals are susceptible to interference by various factors, so the requirements for the surrounding environment are high and the system performance is poor when there are electronic devices or people moving around the user.

### 2.3. Computer Vision-Based Gesture Recognition Technology

At present, computer vision-based gesture recognition techniques are relatively mature and have shown excellent performance with convenience, low cost, and high accuracy. Khaja Shareef et al. [19] implemented a system for deaf gesture recognition based on a deep learning algorithm, which captures the gestures of deaf people through a camera and translates the sign language that is difficult to understand by normal people to provide a simple way of communication. out to provide a simple way of communication. Qazi Mohammad Areeb et al. [20] proposed an RNN-based dynamic gesture recognition system for recognizing Indian sign language, by feeding a small video of sign language to the model, the meaning expressed by the sign language in the video can be derived. Frolova et al. [21] proposed the most probable by extending the longest common subsequence algorithm [22] longest common subsequence algorithm for digital gesture recognition. Stergios Poularakis et al. [23] proposed a complete gesture recognition system based on maximum cosine similarity and a fast nearest neighbor technique, which captures the user’s gesture by a camera and translates the gesture into the corresponding command with simplicity, accuracy, and low complexity. Jubayer Al Mahmudet al. [24] proposed a 3D gesture recognition and adaption system based on Kinect for human-robot interaction. Harshala Gammulle et al. [25] proposed a single-stage continuous gesture recognition framework, called temporal multimodal fusion (TMMF), which enables the detection and classification of multiple gestures in videos through a single model. David González León et al. [26] used a deep camera and a lightweight convolutional neural network (CNN) model for the recognition of input gesture videos with higher accuracy on the test set. Daniel Skomedal Breland et al. [27] proposed a robust gesture recognition system based on high-resolution thermal imaging for the accurate classification of high-resolution gestures by using deep CNNs.

This paper focuses on building a complete gesture recognition system using CV techniques, including gesture coordinate capture, segmentation, representation, and classification. Unlike the direct recognition of captured gesture images in the above work, this paper uses the trajectory points of gesture actions to generate images for gesture recognition, which effectively reduces the model complexity and improves the speed and accuracy of gesture recognition because the images shield the complex features of gestures in 3D space.

## 3. Preliminary and Observation

This section is dedicated to answering the main challenges presented in Section 1, presenting the corresponding solution ideas, and verifying the proposed model efficiency via preliminary experiments. Section 4 describes the details of segmentation and recognition algorithms.

### 3.1. How to Get the User’s Gesture Coordinates?

How to get the gesture coordinates is the first step to performing gesture recognition, to address this challenge, this paper tracks the user’s gesture coordinates in real-time by target detection algorithm.

Some of the existing target detection algorithms are Fast R-CNN (Fast Region-based Convolutional Network) [28], Faster R-CNN(Faster Region-based Convolutional Neural Networks) [29], SSD (Single Shot MultiBox Detector) [30], and YOLO (You Only Look Once). The YOLO algorithm, a typical representative of a single-stage (one-stage) target detection algorithm, has been vastly applied due to its high speed and accuracy rate. YOLOv5 [31] is the latest versions of the YOLO algorithm iteration and is better than YOLOv4 [32] owing to the enhanced speed and accuracy in performance. Hence, YOLOv5 was deployed in this study to capture the coordinates of the user’s gesture trajectory. The gesture-tracking effect is illustrated in Figure 1.

When drawing gestures in a picture with a resolution size of 640*384, a problem might arise if the coordinates differ greatly when different users draw the gesture, thus reducing the comparability between the gesture coordinates captured from different users and the gesture recognition accuracy. After gaining the gesture coordinates, a normalization algorithm was used to limit the gesture coordinates to a similar order of magnitude. Next, the coordinates were connected and drawn in a 2D coordinate system to capture the trajectory picture of the numbers and letters drawn by the user. Figure 2 shows the trajectory images of the numbers and letters drawn based on gesture coordinates.

### 3.2. How to Split Continuous Gestures?

The gestures were typically drawn in a combination of numbers and letters to express certain meanings (e.g., “hello”, “bye”, and “13”). Since the convolutional neural network (CNN) trained in this study could only reckon single numbers or letters, it is crucial to segment continuous gestures into single gestures. The analyzed gesture trajectory data showed that the users did not draw continuous gestures in one breath, as a time pause was observed between each letter or number (the average pause time of each user exceeded 0.25 s for the eight-frame picture). The scanned time interval was used as the dividing line of two adjacent gestures. As the gesture was determined by both x- and *y*-axis, the slope of both axes was considered when determining the dividing line between the two gestures. Figure 3a,b present the trajectory of gesture “147” on the x- and *y*-axis, respectively. The two green dashed lines signify the start and end of gesture “4”. Gesture “4” was drawn using the following three steps: (1) draw a diagonal line, (2) draw a horizontal line, and (3) draw a vertical line. The red dashed line in the figure shows step 2, where the slope is close to zero because the value of the *y*-axis remains constant, but this does not mean that the gesture has stopped. The value of gesture coordinates on the *x*-axis continued changing and the slope was not zero.

Upon considering the slope of the x- and *y*-axis simultaneously, the algorithm complexity increased. To aid the analysis of the algorithm, the values of gesture coordinates from both axes were fused and plotted in 2D form. Figure 4 presents the trajectory of gesture “147” after feature fusion, in which the *y*-axis denotes the fused feature values, x signifies the frame serial number, and the two adjacent red dashed lines from left to right indicate the start and end of a single gesture, respectively.

The feature map portrays the time interval between two adjacent gestures. The eigenvalues of the gesture coordinates during the time interval were smooth (the slope is close to zero). A time window-based gesture segmentation algorithm was built to scan the set of feature points with an average slope less than a specified threshold within the time window. The boundary of the set was used as the sign of the start and end of the gesture. To perform a preliminary validation of the proposed algorithm, four volunteers participated in the experiment by drawing numbers (“13” and “2691”) and words (“hello” and “bye”). Figure 5 illustrates the segmentation effect of the algorithm.

After weighing in the subtle variances in the pause time of each person, the algorithm might have one or two frame sequence number errors when detecting the location of the start and end of a single gesture. However, it did not affect the overall model accuracy.

### 3.3. How to Remove Interference from Surrounding People?

When a user is drawing gestures and other people are walking around, the camera can detect the gestures of the surrounding people. This causes interference with the collected gesture coordinates. Distinguishing the user from other people via a face detection algorithm [33] demands plenty of training data and more network overhead, which adds a burden to the system. To overcome this issue, methods discussed in [34,35] were adhered to, along with the YOLOv5 algorithm that initiates an attention mechanism so that the model can focus on key information in the image and filter undesired information. As the user faced the camera when drawing the gesture, the distance of the hand from the camera was closer than other people, which took up a larger fraction of the picture with clearer features. This distinguished the gestures from other people in the surrounding. Figure 6a,b show the trajectories of the gestures for the letter “a” captured without and with the attention mechanism, respectively, when a person passed behind the experimenter with a clenched fist as an interruption.

We observe that although the YOLOv5 algorithm using the introduced attention mechanism can remove most of the interference well, the error is still inevitable, especially when the user is drawing more complex gesture actions, due to the capture of more frame pictures, there will still be some frames captured the gesture coordinates of other people around, if not removed, it may have an impact on the detection results, for this reason, this paper this proposed gesture coordinate correction algorithm, which contains two parts: (1) Coordinate filtering algorithm based on anchor box size. The algorithm filters out the gestures of interfering persons by comparing the area of the anchor boxes in the prediction result set. Since the user is drawing the gesture facing the camera, the distance of the hand from the camera is closer than other people in the background, and the proportion in the picture is larger, so it can be well distinguished from the gestures of interfering people in the background. (2) Outlier coordinate point removal algorithm. This algorithm mainly solves the problem that in some of the captured gesture pictures, the anchor box size predicted by the target person and the interferer is similar or the YOLOv5 model only detects the interferer gesture due to the ambiguity of the target person’s gesture, which leads to the situation that the coordinate filtering algorithm based on the anchor frame size fails to filter out the interference correctly. By neglecting points where the gesture coordinates in the captured frame at a certain moment change drastically when compared to the previously collected coordinates, the remaining errors could be removed effectively. Figure 6c displays the captured gesture trajectory after combining the YOLOv5 algorithm with the gesture coordinate correction algorithm that introduces the attention mechanism.

### 3.4. How to Recognize Gestures?

How to accurately and quickly recognize the gesture coordinates after segmentation by gesture segmentation module is also an important issue to be considered. To solve this problem, a lightweight convolutional neural network LGR-CNN is constructed in this paper, and the recognition of user gestures is achieved by plotting the segmented gesture coordinates in a two-dimensional coordinate system and feeding them into the network.

## 4. System Design

This section introduces the Air-GR system, followed by a detailed description of the core modules of the system and the algorithms applied in this study.

### 4.1. System Architecture

Figure 7 shows that the Air-GR system comprising of three modules: the gesture trajectory acquisition module, continuous gesture segmentation, and the processing module, as well as the gesture drawing and recognition module.

First, in the gesture trajectory acquisition module, each frame during gesture movement was captured using a Kinect camera. Next, the gesture coordinates in each frame were detected by using the YOLOv5 target detection algorithm, which considered the interference of gestures from other people (multiple coordinates were detected). The interference was removed by using the proposed gesture coordinate correction algorithm. Then, the processed coordinates were inputted into the continuous gesture segmentation and a processing module. To facilitate both observation and analysis, the captured gesture coordinates were normalized and feature fused. The continuous gestures were segmented by using a time window function. In the last module, the processed gesture coordinates were plotted in a 2D coordinate system and sent to the CNN for recognition.

### 4.2. Gesture Track Acquisition Module

#### 4.2.1. The YOLOv5 Algorithm

The model structure of YOLOv5 is composed of four parts: input layer, Backbone, Neck, and Prediction. During the model training process, the input images were first enhanced with Mosaic data in the input layer, and the best anchors of the dataset were obtained by using the K-Means [36] approach. Next, feature extraction was performed using Focus and SPP(Spatial Pyramid Pooling) modules in Backbone. Lastly, the prediction results were gained from the Prediction layer and regressed based on the confidence level.

The loss function of YOLOv5 is composed of bounding box loss $L_{\mathrm{box}}$, confidence loss $L_{\mathrm{obj}}$, and classification loss composition $L_{\mathrm{cls}}$, as expressed in Equation (1).
(1)$$L_{yolov5}=L_{box}+L_{obj}+L_{cls}$$

Classification loss $L_{\mathrm{cls}}$ and confidence loss $L_{\mathrm{obj}}$ were calculated by using the binary cross-entropy loss function, as presented in Equation (2):(2)$$C=-\frac{1}{n}\underset{x}{\sum }\left[y\ln a+\left(1-y\right)\ln\left(1-a\right)\right]\;$$
where x is the sample, y denotes the label, a  refers to the predicted output, and n represents the total number of samples. The GIOU (Generalized-IOU) algorithm [37], which was applied to determine the bounding box loss, can overcome the setbacks of IOU (Intersection over Union) [38] while making full use of the advantages of IOU. The loss function of GIOU is presented in Equation (3).
(3)$$GIOU=1-GIOU=1-\frac{C-B}{C}\;$$
where B is the union of the ground truth box and prediction box, while C denotes the minimum bounding rectangle of the prediction box and real box. The GIOU loss function is based on the IOU loss function with the consideration of another factor. The GIOU not only weighs in the overlapping area between the detection box and the target box, but also concentrates on the non-overlapping part of other regions to reflect the overlap between the prediction box and the target box. This increases the measurement of the intersection scale between the two.

#### 4.2.2. YOLOv5 Algorithm with the Introduction of the CBAM Attention Module

To attain better detection outcomes, the CBAM (Cost Benefit Analysis Method) [39] attention mechanism was deployed in this study by incorporating the CBAM module after the activation function in the convolution module of the YOLOv5 model. The CBAM has two sequential sub-modules of channel attention (Channel Attention Module (CAM)) and spatial attention (Spatial Attention Module (SAM)). Upon inputting the gesture feature map, it first entered the CAM based on the width and height of the feature map for both GAP (Global Average Pooling) and GMP (Global Max Pooling). The attention weight of the channel was retrieved using MLP (MultiLayer Perceptron), while the normalized attention weight, which was gained via the Sigmoid function, was weighted to the original input feature map by multiplication. This step completed the recalibration of channel attention to original features. The formula is presented in Equation (4).
(4)$$\begin{array}{ll} M_{C}\left(F\right) & =\sigma \left(MLP\left(AvgPool\left(F\right)\right)+MLP\left(MaxPool\left(F\right)\right)\right)\\ & =\sigma \left(W_{1}\left(W_{0}\left(F_{avg}^{c}\right)\right)+W_{1}\left(W_{0}\left(F_{max}^{c}\right)\right)\right)\; \end{array}$$
where σ is sigmoid function, whereas $W_{0}\in R^{C/r\times C}$ and $W_{1}\in R^{C\times C/r}$ are MLP weights.

To gain attentional features in the spatial dimension, the feature map output from channel attention was subjected to both global maximum pooling and global average pooling based on the width and height of the feature map. The feature dimension was converted from w×h to 1×1. Next, the feature map dimension was reduced after a 7×7 convolution kernel and Relu activation function, which was later raised to the original dimension after convolution. Lastly, the feature map that was normalized via the Sigmoid activation function had been merged with the feature map generated by channel attention. This step completed the rescaling of the feature map in both spatial and channel dimensions, as expressed in Equation (5).
(5)$$M_{s}\left(F\right)=\sigma \left(f^{7\times 7}\left(\left[AvgPool\left(F\right);MaxPool\left(F\right)\right]\right)\right)=\sigma \left(f^{7\times 7}\left(\left[F_{avg}^{s};F_{max}^{s}\right]\right)\right)\;$$
where $f^{7\times 7}$ is a convolutional operation with a convolutional kernel size of 7×7.

### 4.3. Gesture Coordinate Correction Algorithm

The gesture coordinate correction algorithm consists of two parts: (1) Coordinate filtering algorithm based on anchor box size. (2) Outlier coordinate point removal algorithm.

#### 4.3.1. Coordinate Filtering Algorithm Based on the Anchor Box Size

The output of the YOLOv5 algorithm includes the size of the anchor box of the target object in addition to the coordinates of the target object, and the data format of the prediction result Pre is shown in Equation (6).
(6)$$Pre=\left[label\_index,x,y,w\_anchor,h\_anchor\right]\;$$
where label_index is the index of the predicted label name in the label array, x, y denote the *x*-axis and *y*-axis coordinates of the anchor box center point, and w_anchor, h_anchor denote the width and height of the predicted anchor box, respectively.

By using the width and height of the anchor box we can calculate the corresponding area and select the coordinates corresponding to the anchor box with the largest area from the prediction result set R of the image as the output of that image based on the calculation result. The formula is presented in Equation (7):(7)$$R=\left\{Pre_{1},Pre_{2},\cdots ,Pre_{n}\right\}\;S_{i}=Pre_{i}^{w\_anchor}\times Pre_{i}^{h\_anchor}\;S_{max}=max\left(S_{1},S_{2},\cdots ,S_{n}\right)\;$$
where R denotes the set of prediction results, which may contain multiple prediction results Pre. ${\mathrm{Pre}}_{i}^{w},\;{\mathrm{Pre}}_{i}^{h}$ and ${\;S}_{i}$ denote the width and height of the anchor box of the ith prediction result and the corresponding area, respectively, and $S_{\max}$ denotes the anchor box with the largest area in the result set, and the coordinates of the center point of this anchor box are the final output results of the corresponding image.

#### 4.3.2. Outlier Coordinate Point Removal Algorithm

To describe the outlier coordinate point removal algorithm, a temporal encoding model is defined in this study. The expression of gesture V (V is a single gesture) that has n coordinates P is given in Equation (8).
(8)$$V=\left\{P_{1},P_{2},P_{3},\cdots ,P_{n}\right\}\;P_{i}=\left(x_{i},\;y_{i}\right)\;$$

Since the gesture coordinates P were varied uniformly with time, the Cartesian distance between gesture coordinates $P_{i}$ captured in any frame i, and gesture coordinates $P_{i-1}$ and $P_{i+1}$ captured in adjacent frames i−1 and i+1 must be lower than the threshold δ. Coordinate points bigger than threshold δ were discarded. The related formula is expressed in Equation (9):(9)$$V=\left\{\cdots ,P_{i-1},P_{i},P_{i+1},\cdots \right\}\;\sqrt{{\left(P_{i}\left(x\right)-P_{i-1}\left(x\right)\right)}^{2}+{\left(P_{i}\left(y\right)-P_{i-1}\left(y\right)\right)}^{2}}<\delta \;\sqrt{{\left(P_{i}\left(x\right)-P_{i+1}\left(x\right)\right)}^{2}+{\left(P_{i}\left(y\right)-P_{i+1}\left(y\right)\right)}^{2}}<\delta$$

### 4.4. Continuous Gesture Segmentation and Processing Module

#### 4.4.1. Gesture Coordinates Normalization

Notably, PN is the normalized coordinates, while VN is the set of coordinates related to gesture V after normalization. The formula is presented in Equation (10):(10)$$PN_{i}=\left(\frac{P_{i}\left(x\right)}{w},\frac{P_{i}\left(y\right)}{h}\right)\;VN=\left\{PN_{1},PN_{2},PN_{3},\cdots ,PN_{n}\right\}\;$$
where $P_{i}\left(x\right)$ refers to the original *x*-axis coordinate, $P_{i}\left(y\right)$ is the original *y*-axis coordinate, while h and w are pixel values of the height and width of the image, respectively.

#### 4.4.2. Continuous Gesture Segmentation Algorithm

In describing the continuous gestures, G signifies the set of gestures obtained by the gesture acquisition module. The expression of G is as follows:(11)$$G=\left\{VN_{1},VN_{2},VN_{3},\cdots ,VN_{m}\right\}=\left\{PN_{1},PN_{2},PN_{3},\cdots ,PN_{mn}\right\}$$

Since G might contain m gestures VN, the set of gestures G was segmented to detect the location where a single gesture ended. Referring to the experimental observations depicted in Section 3, a time window function T of length 2t was constructed to scan the values of gesture coordinate set $\left\{{\mathrm{PN}}_{i-t},\cdots ,{\mathrm{PN}}_{i-1},{\mathrm{PN}}_{i},{\mathrm{PN}}_{i+1},\cdots ,{\mathrm{PN}}_{i+t}\right\}$ in x- and *y*-axis in time window T. As for the algorithm analysis, the axes values of the normalized gesture coordinates PN were feature fused. After the fusion, the value of each feature coordinate point ${\mathrm{Feature}}_{i}$ is as given in Equation (12).
(12)$$\;Feature_{i}=\frac{1}{2}PN_{i}\left(x\right)+\frac{1}{2}PN_{i}\left(y\right)\;$$

The set of characteristic coordinate points $G_{\mathrm{feat}h\mathrm{er}}$ is expressed in Equation (13).
(13)$$\;G_{feather}=\left\{Feature_{1},Feature_{2},Feature_{3},\cdots ,Feature_{mn}\right\}\;$$
when using the set of feature coordinate points $G_{\mathrm{feat}h\mathrm{er}}$, it is merely necessary to scan the set $G_{\mathrm{feat}h\mathrm{er}}$ based on the defined time window T, besides calculating the average slope of the set ${\mathrm{Feature}}_{i}$ centered on feature point $\left\{{\mathrm{Feature}}_{i-t},\cdots ,{\mathrm{Feature}}_{i},\cdots ,{\mathrm{Feature}}_{i+t}\right\}$ in each time window T. Upon choosing the set of feature coordinate points with the smallest average slope, the boundaries of the start and the end of a single gesture were determined. The variance was used in this study to calculate the slope in time window T. A sliding window segmentation method based on variance is proposed in this study, where ${\;k}_{i}$ is the slope of the whole time window T. The retrieved value denotes the variance centered on the feature point ${\mathrm{Feature}}_{i}$ in time window T. Next, the slope magnitude in the current time window and the threshold μ were determined. A slope exceeding threshold μ signifies that the gesture action in the time window has not converged to a stable state. When the slope is below the threshold μ, it means that the gesture is stable (the user finished drawing a single gesture) and the left boundary of the time window denotes the end position of a single gesture, as represented by l, and the slope is signified by $k_{p}$. As it is impossible to determine the time when the next gesture begins, it is crucial to continue to search backward. When the slope of the subsequent time window exceeds μ, it denotes that the user has begun drawing the second gesture in this time window, with the right boundary of the time window representing the start position of the next gesture, which is denoted by r and the slope recorded as $k_{q}$. Next, the boundary of the gesture was determined based on subscripts p and q, where p−t is the subscript of the frame and the final gesture ends, while q+t is the subscript of the frame where the next gesture begins. The subscripts are represented by l and r, respectively. The specific formula is given below.
(14)$$\;k_{i}=\frac{1}{2t}\sum_{j=i-t}^{i+t}{\left(Feature_{j}-Feature_{i}\right)}^{2}\;k_{p},k_{q}\leq \mu \;\;l=p-t,\;r=q+t\;$$

### 4.5. LGR-CNN Module

In this module, to recognize the drawn gesture trajectory map quickly and accurately, this paper improves the structure of the traditional convolutional neural network and builds a lightweight convolutional neural network LGR-CNN (Lightweight Gesture Recognition CNN), the network structure of LGR-CNN is shown in Figure 8. Firstly, for the problem of large image size, an odd-even sampling module is proposed to sample the original image by two MaxPool of size 1×1, which can better preserve the key information of the image. Secondly, Ghost convolution [40] is used instead of most conventional convolution operations, which can effectively reduce the number of model parameters while ensuring accuracy. Finally, the gesture recognition accuracy is further improved by introducing the ECA attention module [41] in the model to extract the dependencies between channels.

#### 4.5.1. Parity Sampling Module

Although the drawn gesture trajectory map is simpler than the gesture picture containing complex features in 3D, the gesture trajectory curve only accounts for a small part of the picture, resulting in a large amount of redundant information in the picture, which will increase the computational complexity of the model if it is directly put into the network for training. The common method is to process the image by Resize or convolution/pooling operation (downsampling), however, simple Resize will make the image pixel loss, while convolution/pooling operation is often used after the convolution layer, generally first through the convolution layer for feature fusion, and then downsample the output features, if the input layer directly downsamples the original image, it may filter the useful information. To solve the above problem, a parity sampling module is constructed in this paper, as shown in Figure 9, in which the original input is first divided into two branches, one of which is oddly sampled by MaxPool of size 1 and stride 2. In the other branch, the image is first filled by Padding operation, and then evenly sampled by MaxPool of size 1 and stride 2. Since the size of the image after even sampling will have one more row and one more column with a 0-pixel value, it needs to be cropped by Crop operation to make the size of the image after odd and even sampling consistent. Finally, the output feature map is summed with the first branch by Concat. Since the Maxpool of size 1*1 does not change the original features of the image, it can better retain the key information of the original image and achieve the effect of reducing the dimensionality of the image at the same time.

#### 4.5.2. Ghost Module

The conventional feature extraction approach is to use multiple convolutional kernels to perform convolutional mapping operations on all channels in the input feature map. However, as the number of network layers deepens, stacking a large number of convolutional layers requires a huge number of parameters and computational effort, and also generates many rich or even redundant feature maps. Therefore, this paper uses the Ghost module to replace most of the regular convolution operations. The Ghost convolution operation is shown in Figure 10, which first uses a small number of regular convolution kernels to extract features from the input feature map, then further performs more inexpensive linear transformation operations on this part of the feature map, and finally generates the final feature map by the Concat operation.

Assuming that the data input to the Ghost module is $X\in R^{c\times h\times w}$, the number of channels is c, and the height and width are h and w, the m feature maps generated by the convolution layer can be expressed as
(15)Y=W⊗X+B
where ⊗ denotes the convolution operation, B is the bias term, which is negligible, $Y\in R^{m\times h^{\prime }\times w^{\prime }}$ is the output feature map, $h^{\prime }$ and $w^{\prime }$ are the width and height of the output feature map, $W\in R^{c\times k\times k\times m}$ is the convolution kernel weight matrix, and k×k is the convolution kernel size.

Afterward, on the feature map Y, the s Ghost feature maps are generated using simple linear operations according to Equation (16).
(16)$$y_{ij}=\Phi_{i,j}\left(y_{i}\right),\;\forall i=1,\cdots ,m,\;j=1,\cdots ,s\;$$
where ${\;y}_{i}$ denotes the i-th feature map in Y and $y_{\mathrm{ij}}$ denotes the j-th Ghost feature map generated from the i-th feature map by linear operation $\Phi_{\mathrm{ij}}$.

#### 4.5.3. ECA Module

Unlike the CBAM attention mechanism used in the gesture detection algorithm, for the picture classification task, the channel information is more important because it does not need to pay much attention to the position of the drawn gesture trajectory in the picture. Based on this consideration, the ECA attention module is added behind the last three convolutional layers of LGR-CNN, and the network structure is shown in Figure 11.

The ECA attention module mainly consists of three operations, GAP (Global average pooling), Conv1d, and Scale. Firstly, the input feature map is transformed into an output feature map of 1×1×c size by the global average pooling operation without dimensionality reduction to obtain the global information. After that, the weights of each channel are calculated using Conv1d, and the number of weights is determined by the convolutional kernel size k, which should vary in size for a different number of channels c. Therefore, ECA proposes an adaptive method to determine the size of the one-dimensional convolutional kernel with the formula:(17)$$c=2^{\gamma \cdot k-b}$$

Therefore, when the number of channels c is determined, the adaptive convolution kernel size k can be calculated according to Equation (18).
(18)$$k={\left|\frac{\log_{2}\left(c\right)}{\gamma }+\frac{b}{\gamma }\right|}_{odd}\;$$
where γ and b generally take values of 2 and 1, and ${\left|x\right|}_{\mathrm{odd}}$ denotes the nearest odd number to x. Finally, the weights are weighted to each channel feature by the Scale operation.

## 5. Experiments and Results

This section presents the implementation of the Air-GR system and the evaluation of its performance through a series of extensive experiments.

Hardware: The hardware used in this study included a Kinect camera with a 30-frame rate and a Lenovo R7000p computer for data reception and processing.

Dataset: Eighteen volunteers (6 female and 12 male adults) participated in the experiment. In total, 6860 data samples were collected, of which 2100 referred to hand images that were manually labeled in accordance with the Visual Object Classes (VOC) dataset format. These images were applied to train the enhanced YOLOv5 model. The remaining 4760 data samples comprised hand gesture data, including numbers 0–9, letters a-z, as well as the consecutive numbers and letters, which were deployed to train and test the CNN.

Experimental environment: The experiments were performed in an empty classroom (area: ~9 m * 6 m (differed from the training data setting)). The classroom where the experiment was held is presented in Figure 12. During the experiment, the user sat before the desk where the Kinect camera was placed to draw gestures. The Kinect camera transmitted each captured frame to the PC via Ethernet for gesture recognition.

Metrics: To elaborate on the performance of the Air-GR system, accuracy rate (ACC) and intersection ratio (IOU) were employed as the evaluation metric.

The ACC refers to the measurement of the likelihood of correctly recognizing user gesture actions. This was calculated as follows:(19)$$ACC=\frac{TP+TN}{TP+FP+FN+TN}$$
where TP and FP are samples predicted by the model as positive cases but correctly and incorrectly predicted, respectively. Meanwhile, TN and FN refer to samples predicted by the model as negative cases but correctly and incorrectly predicted, respectively.

IOU is used to evaluate the overlap between the anchor box output by the model and the real anchor box, and a larger value indicates that the algorithm predicts better, and the mathematical definition of IOU is as follows.
(20)$$IOU=\frac{\left(A\cap B\right)}{\left(A\cup B\right)}\;$$
where A is the prediction box and B is the ground truth box.

### 5.1. Gesture Detection Accuracy

In order to verify the effectiveness of the proposed gesture coordinate correction algorithm, a comparison experiment is designed in this paper. The traditional YOLOv5 algorithm is denoted as YOLOv5; the YOLOv5 algorithm after introducing the CBAM attention mechanism is denoted as YOLOv5-CBAM; and the algorithm after combining YOLOv5-CBAM with gesture coordinate correction is denoted as YOLOv5-CBAM-GCC. the above three algorithms are trained using the same data set, and to test the trained models To test the trained model, we invited 10 volunteers to participate in the experiment and divided them into 5 groups. The first group of volunteers drew the gestures without interference factors. The second group of volunteers drew the gesture with a person passing behind them as a distractor, and so on. Each volunteer drew the gesture for at least 3 s (the Kinect camera captured about 30 frames per second). After the drawing was completed, the number of pictures of gestures captured by Kinect that captured only the target user’s gestures after detection by the YOLOv5 model was calculated as the proportion of all pictures, as the detection accuracy of the model. The experimental results are shown in Table 1 and Table 2. Since the presence or absence of interfering factors has little effect on the IOU, in Table 2 we only consider the IOU values of the model in scenes without interfering factors.

By analyzing the results in Table 1 and Table 2, we observe that the average target person gesture detection accuracy of all three algorithms reaches more than 97% when there are no interfering person gestures in the gesture picture. However, since the latter two algorithms use the YOLOv5 algorithm that introduces an attention mechanism, the average accuracy and IOU values are higher than those of the traditional YOLOv5 algorithm. Since there are no interfering persons, the gesture coordinate correction algorithm provides no help to the model, resulting in the same average accuracy and IOU values for YOLOv5-CBAM and YOLOv5-CBAM-GCC.

When there are interferer gestures in the gesture picture, the gesture detection accuracies of the three algorithms show a more obvious gap. First, for the traditional YOLOv5 algorithm, with the increase of the number of interferers, its target person gesture detection accuracy shows a significant decrease, especially when the number of interferers is 4, its accuracy rate is the lowest, only 64.7%. Second, for the YOLOv5-CBAM algorithm, although it can remove most of the interference well and can improve 13.1% on the basis of the traditional YOLOv5 algorithm when the number of interferers is 4, the detection accuracy of 77.6% still cannot meet the needs of practical applications. Fortunately, by combining the YOLOv5-CBAM algorithm and the gesture coordinate correction algorithm, the model accuracy can be effectively improved. As can be seen from the results in the table, the YOLOv5-CBAM-GCC algorithm provides 98.5%, 98.3%, 98.7%, and 98.9% target person gesture detection accuracies for four scenarios with different degrees of interference factors, respectively. Although there are still errors, by analyzing the detected gesture pictures, we observe that the errors are mainly due to the existence of certain frames in which the user’s gestures are blurred during the drawing process, resulting in YOLOv5 failing to recognize the gestures in the pictures, but considering that the gesture is a continuous set of coordinate points, the loss of a small number of gesture coordinates does not affect the drawing of the gesture trajectory map.

### 5.2. Accuracy of Continuous Gesture Segmentation

To assess the model accuracy for the segmentation of gestures at varied lengths, 11 volunteers participated in the experiment. Each volunteer drew any combination of English/number of lengths 2, 3, 4, 5, and 6; 10 times for each combination. As a result, 550 samples were captured as test data. Next, a confusion matrix was used to test the segmentation accuracy of the model. Figure 13 illustrates the segmentation accuracy for five different lengths of gestures. The outcomes revealed that the segmentation accuracy decreased slightly as the length of the gesture increased. Words are more complex and take a longer time to draw than numbers even for similar lengths, thus leading to lower accuracy of word segmentation than numbers. After statistically analyzing 3000 words typically used on the website, the average length of commonly used words is six. For words with a length less than or equal to six, the model scored 97.48% accuracy, which meets most of the needs in real life.

### 5.3. Ablation Study

To verify the improvement of model performance by the proposed parity sampling module in this paper, ablation experiments are designed. The model using the parity module is denoted as LGR-CNN; the model using MaxPooling for downsampling is denoted as LGR-CNN-MaxPooling; the model using the convolution operation for downsampling is denoted as LGR-CNN-Conv; and the model after the simple Resize operation of the input image is denoted as LGR-CNN-Resize. ablation experiments The accuracy curves on the test set are shown in Figure 14.

It can be seen that the simple Resize operation causes image pixel loss, resulting in the network accuracy curve only stabilizing around 87% after a tortuous rise. Although downsampling by MaxPooling and convolution can improve the recognition accuracy to some extent, it still cannot preserve the image feature information well if added directly behind the input layer of the network, resulting in the accuracy of both only 94.3 and 92.1%. In contrast, the odd-even sampling module proposed in this paper can better preserve the key information of the original image while reducing its dimensionality because it does not change the original features of the image, and its accuracy reaches 98.2%.

### 5.4. Comparison of Different Image Classification Models

To evaluate the gesture trajectory recognition model LGR-CNN proposed in this paper, three models, ResNet18, GooLeNet, and VGG16, are selected to train on the gesture trajectory image dataset collected in this paper, and the accuracy rate on the test set is recorded. The number of training sets is 2772, and the number of test sets is 1188. the training result curve is shown in Figure 15, and it can be seen that the LGR-CNN model proposed in this paper gradually smooths out when the epoch is greater than 60, and the accuracy rate reaches 98.2%. Although the number of parameters included in VGG16 is very large, it is easy to cause overfitting because it is mainly concentrated in the fully connected layer, resulting in the accuracy of its test set not being higher than 85%. resNet18 model can solve the gradient disappearance or explosion problem well by introducing the residual structure, but the final correct rate is only about 86% because the actual number of network layers is not very deep; GoogLeNet is a network with deeper network depth, and it overcomes the gradient disappearance problem during training due to the inclusion of auxiliary classifier in the network, and performs well in the present set, reaching 93.7% accuracy, but the model training convergence speed is not as fast as LGR-CNN. The experiments demonstrate that the proposed LGR-CNN has higher recognition accuracy and faster convergence speed for the gesture trajectory image classification task.

### 5.5. Overall Accuracy of Air-GR System

To assess the overall accuracy of the Air-GR system, seven volunteers participated in the experiment. Each volunteer independently decided on the length of the gesture to be drawn. In total, 315 measured samples were gathered, inclusive of continuous numbers and English letters. The experimental outcomes are illustrated in Figure 16. The accuracy scores of the seven volunteers were 93.33%, 97.78%, 93.33%, 95.56%, 95.56%, 93.33%, and 97.78%, whereas the overall accuracy rate decreased when compared with that of a single gesture. This outcome is ascribed to the rhythm of the gestures made by the volunteers influenced by emotions and other factors. While drawing continuous gestures, some consecutive English letters or numbers were drawn too fast and this affected the segmentation process. As observed, the model achieved 95.24% accuracy in continuous gesture recognition.

### 5.6. Comparison of Different Gesture Recognition Algorithms

To demonstrate the effectiveness of the Air-GR proposed in this paper, this paper is compared with some of the latest gesture recognition algorithms, such as literature [42], literature [43], literature [44], literature [45], and literature [46] (see Table 3). The Air-GR system proposed in this paper can perform fast and effective recognition of 26 English letters and 10 digits. In addition, since Air-GR implements segmentation of consecutive gestures, Air-GR can also perform effective recognition for any combination of 26 English letters and 10 digits with a length of less than 7, and its recognition accuracy reaches 95.24%. The literature [42] proposed a human gesture recognition system based on the improved YOLOV3, which has a gesture recognition accuracy of about 90%, which is lower than that of Air-GR. The literature [43] achieved the recognition of 80 Arabic sign languages by connecting a 3D CNN skeleton network to a 2D point convolutional network, but the system could only recognize images with 89.62% accuracy. In the literature [44], a dynamic gesture recognition method based on improved OpenPose was proposed to achieve the recognition of 10 dynamic gestures with 92.4% accuracy, which is lower than the system designed in this paper, and the system needs to consider the information of the joints of the user’s hand, which increases the training amount of the system compared to only considering the hand position. The literature [45] proposed a gesture recognition system based on improved ResNet-50 and MediaPipe with an accuracy of 98.5%, but the system only supports the recognition of seven common gestures, which is much lower than our model, and the system cannot segment continuous gestures, which cannot meet the practical application requirements. The literature [46] proposed a gesture recognition system based on MediaPipe and LSTM, which achieved the recognition of 10 common gestures with 90% accuracy by tracking the skeletal points of the user’s hand, and its recognition quantity and accuracy were lower than Air-GR. the above comparison results proved the effectiveness of the Air-GR model proposed in this paper.

## 6. Discussion

The Air-GR captures the coordinates of the user’s gesture trajectory by a camera and feeds them into the constructed LGR-CNN to achieve recognition of airborne handwritten characters, which greatly reduces the model training complexity and supports segmentation and recognition of continuous gestures. However, given the setbacks of the existing conditions, the use of Air-GR in real life still poses a great challenge. First, more gesture data must be captured to meet the demand of practical applications. In future endeavors, a small dataset may be used for efficient recognition.

Second, the Air-GR model has a large granularity of gesture recognition, thus demanding its users to face the camera and draw gestures by swinging their arms—denoting multiple inconveniences for practical application. Thus, future work should improve the experiments to achieve finer granularity of gesture recognition by tracking the finger movements of the user.

Finally, despite the high accuracy score yielded by the Air-GR system for recognizing numbers and English words, its performance was affected by complex gestures (e.g., Chinese characters). Therefore, in the subsequent experiments, we will focus on improving the recognition strategy and designing a more excellent gesture recognition method. Besides optimizing the model proposed in this study, the Air-GR could be applied to broader aspects, such as Chinese character recognition, Japanese language recognition, etc. Clearly, the proposed Air-GR system can offer significant contributions to the field of gesture recognition.

## 7. Conclusions

In this paper, we propose an aerial handwritten character recognition system Air-GR based on coordinate correction YOLOv5 algorithm and lightweight convolutional neural network LGR-CNN and describe the design of each module of the model. Unlike direct recognition of captured gesture images, we use the trajectory points of gesture actions to generate images for gesture recognition. The gesture coordinates are detected by using a YoloV5 neural network that introduces an attention mechanism, and a gesture coordinate correction algorithm is proposed to solve the problem of removing human interference. After that, to realize the segmentation of continuous gestures, this paper proposes a time window-based gesture segmentation algorithm, which can obtain the beginning and end of a single gesture. Finally, the continuous gesture trajectory is split into single gestures according to the segmented location coordinates, which are plotted in two-dimensional coordinates and input to the constructed lightweight convolutional neural network LGR-CNN for recognition, which was 13.2%, 12.2%, and 4.5% more accurate than VGG16, ResNet, and GoogLeNet mainstream networks, respectively, for the gesture trajectory picture classification task. A large number of experiments have demonstrated that Air-GR can perform fast and effective recognition of any combination of 26 English letters and numbers with lengths of less than 7, and its recognition accuracy reaches 95.24%.

## Figures and Tables

**Figure 1 sensors-23-01464-f001:**
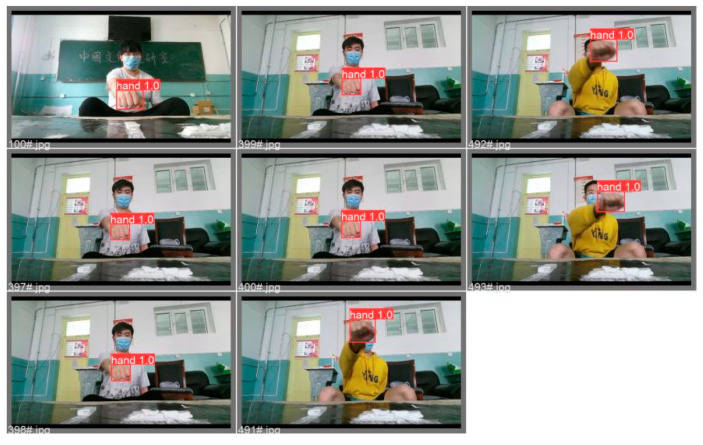
Gesture tracking rendering.

**Figure 2 sensors-23-01464-f002:**
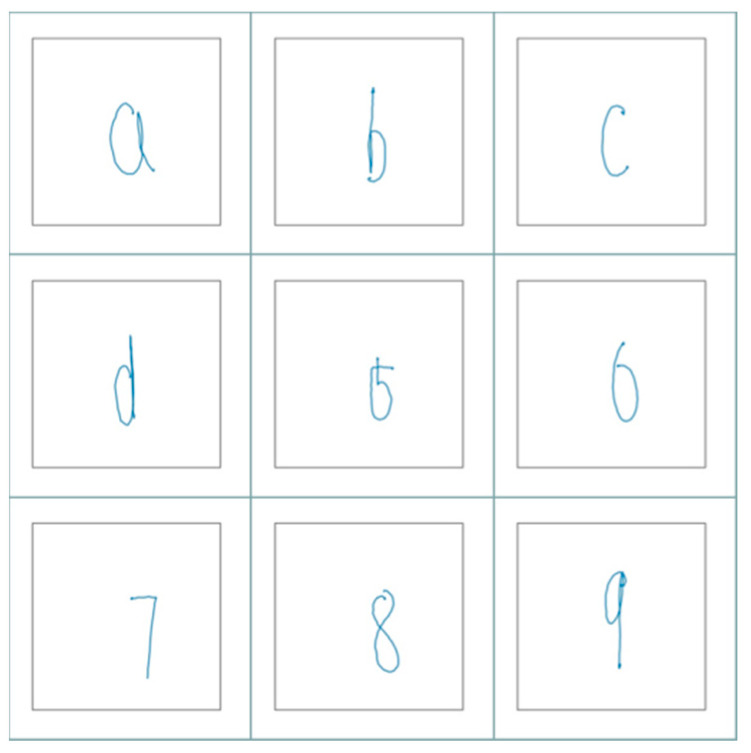
Track diagram of numbers and English letters.

**Figure 3 sensors-23-01464-f003:**
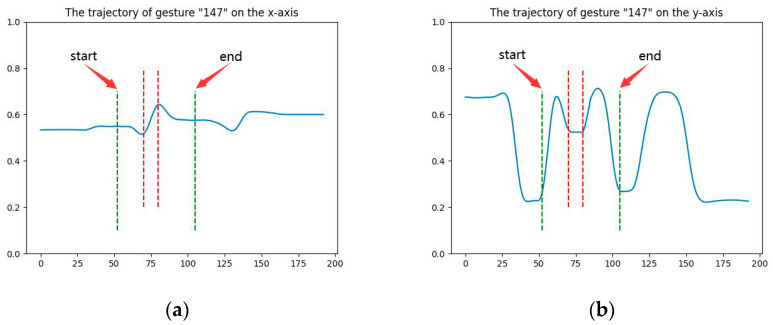
The trajectory of gesture “147” on (**a**) the *x*-axis and (**b**) the *y*-axis.

**Figure 4 sensors-23-01464-f004:**
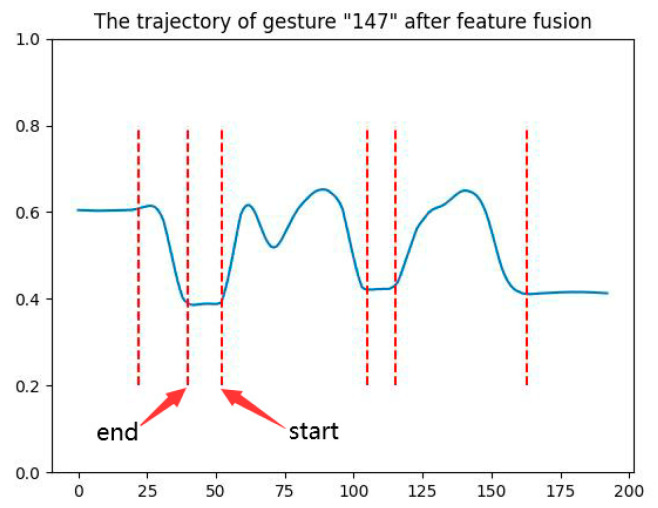
The trajectory of gesture “147” after feature fusion.

**Figure 5 sensors-23-01464-f005:**
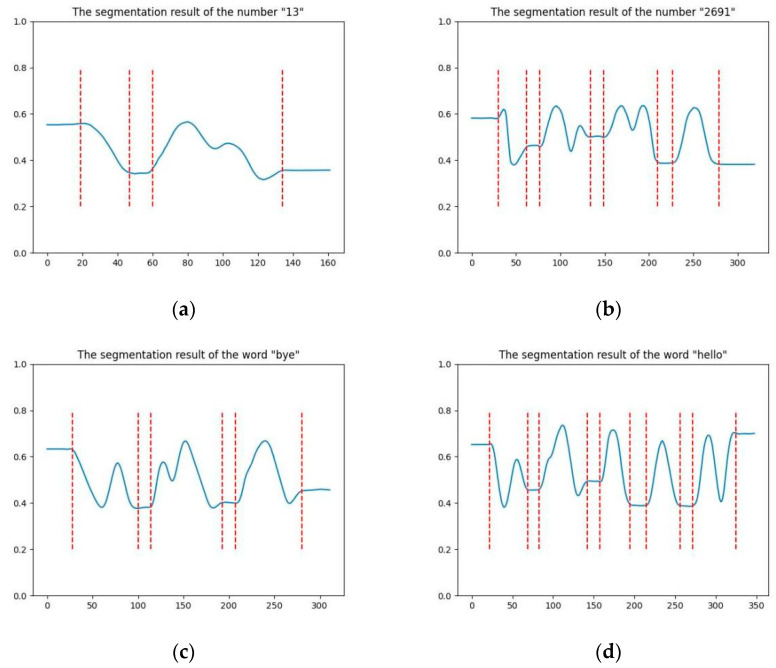
Consecutive gesture segmentation rendering: (**a**,**b**) are the segmentation diagram of numbers “13” and “2691”, and (**c**,**d**) are the segmentation diagram of words “bye” and “hello”.

**Figure 6 sensors-23-01464-f006:**
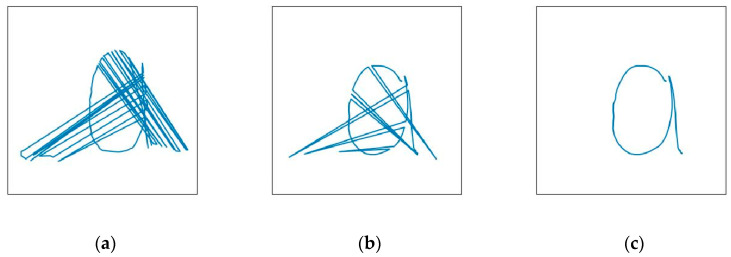
Gesture trajectories of the letter “a” under different algorithms. (**a**–**c**) denote the gesture trajectories captured by the conventional YOLOv5, the YOLOv5 with the attention mechanism introduced, and the gesture coordinate correction algorithm, respectively.

**Figure 7 sensors-23-01464-f007:**
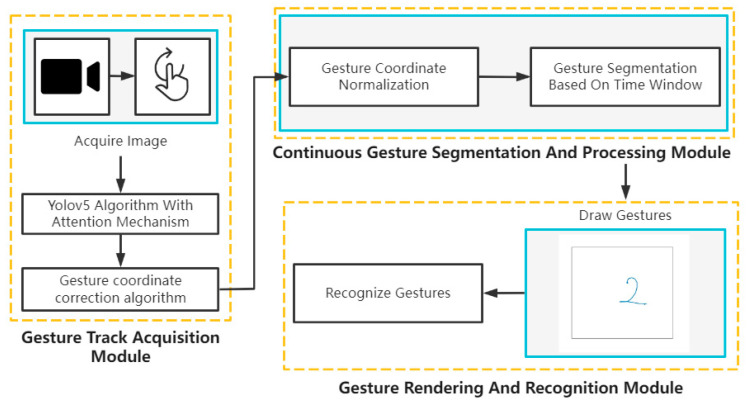
Air-GR System.

**Figure 8 sensors-23-01464-f008:**
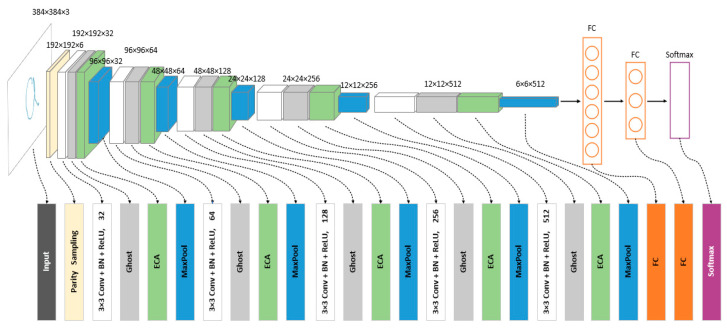
The LGR-CNN Network structure.

**Figure 9 sensors-23-01464-f009:**
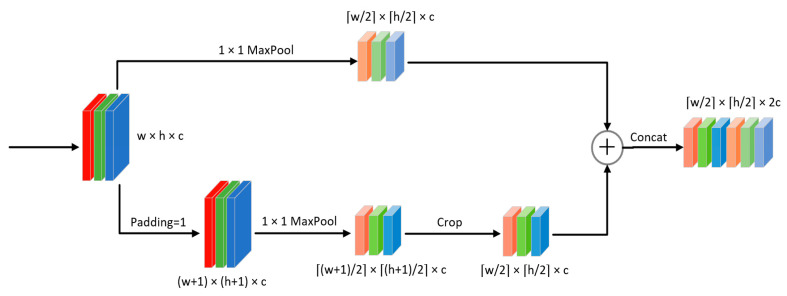
The parity sampling module structure.

**Figure 10 sensors-23-01464-f010:**
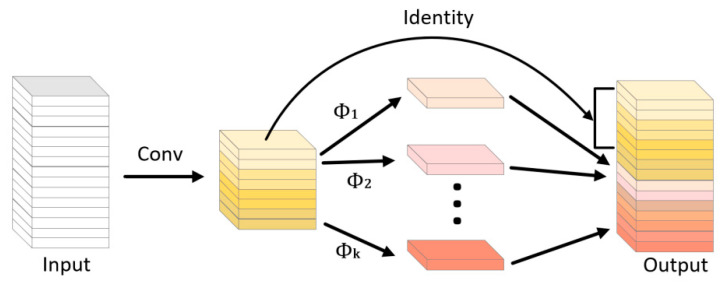
The ghost module structure.

**Figure 11 sensors-23-01464-f011:**
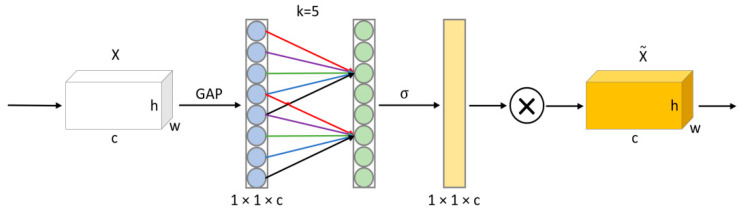
The ECA module structure.

**Figure 12 sensors-23-01464-f012:**
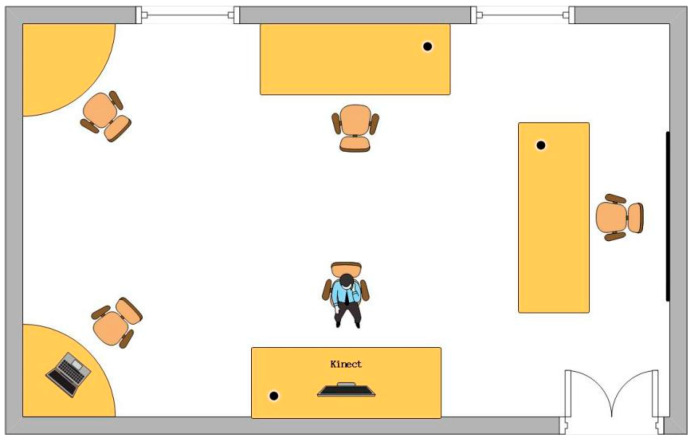
Classroom plan.

**Figure 13 sensors-23-01464-f013:**
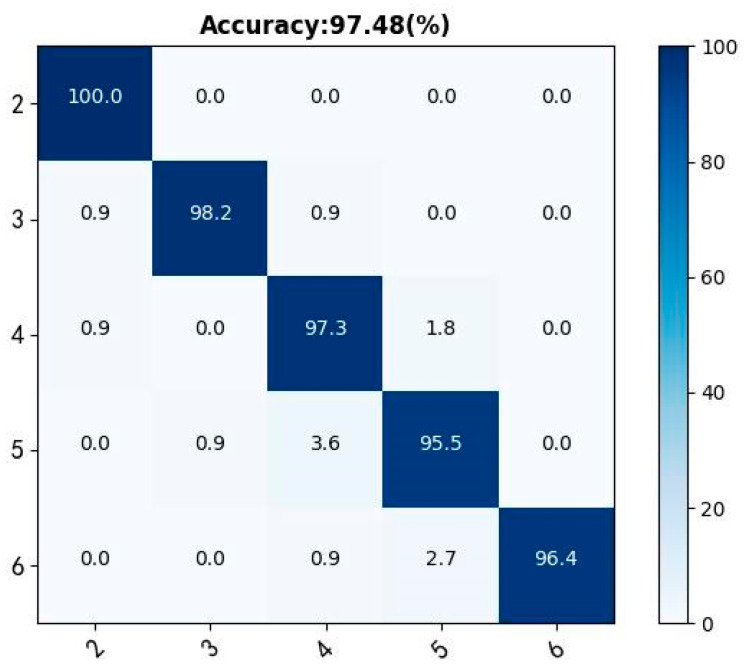
Confusion matrix of different gesture lengths.

**Figure 14 sensors-23-01464-f014:**
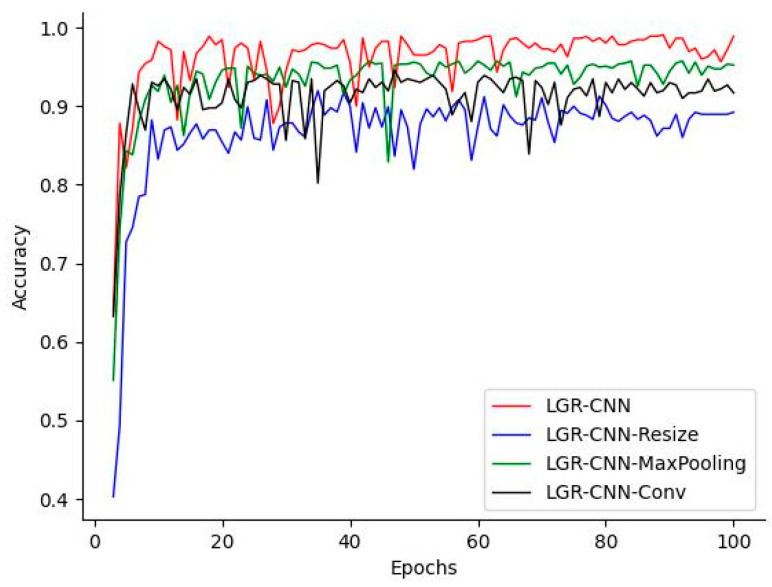
Accuracy curve of ablation experiment.

**Figure 15 sensors-23-01464-f015:**
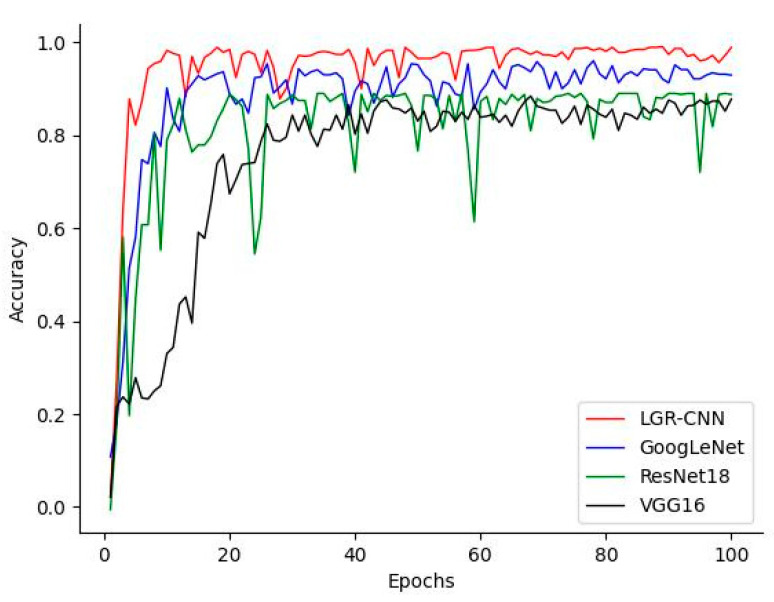
Accuracy of different classification models.

**Figure 16 sensors-23-01464-f016:**
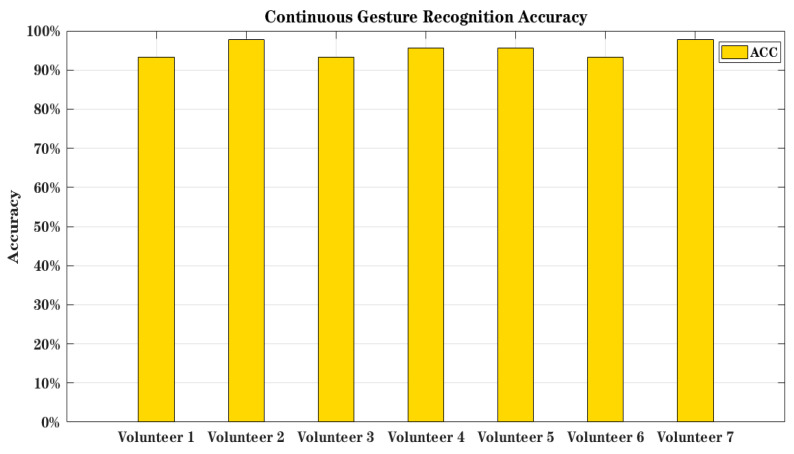
Overall accuracy of Air-GR system.

**Table 1 sensors-23-01464-t001:** Comparison of the average accuracy of different gesture detection algorithms.

	YOLOv5	YOLOv5-CBAM	YOLOv5-CBAM-GCC
0 interfering person	97.3%	99.2%	99.2%
1 interfering person	89.3%	94.3%	98.5%
2 interfering person	81.7%	90.6%	98.3%
3 interfering person	73.4%	82.4%	98.7%
4 interfering person	64.5%	77.6%	98.9%

**Table 2 sensors-23-01464-t002:** Comparison of the average IOU of different gesture detection algorithms.

	YOLOv5	YOLOv5-CBAM	YOLOv5-CBAM-GCC
0 interfering person	94.3%	97.6%	97.6%

**Table 3 sensors-23-01464-t003:** Comparison of different gesture recognition algorithms.

Method	Static/Dynamic	Number	Isolated/Continous	Accuracy
YOLOv3-based [42]	Static	9	Isolated	90%
3DCNN+2DCNN [43]	Dynamic	80	Isolated	89.62%
OpenPose-based [44]	Dynamic	10	Isolated	92.4%
ResNet+MediaPipe [45]	Dynamic	7	Isolated	98.5%
LSTM+MediaPipe [46]	Dynamic	10	Isolated	90%
Air-GR	Dynamic	36	Continuous	95.24%

## Data Availability

Not applicable.

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
