# Peer review of "Air-GR: An Over-the-Air Handwritten Character Recognition System Based on Coordinate Correction YOLOv5 Algorithm and LGR-CNN"

_sensors, 2023, doi:10.3390/s23031464_

Round 1
Reviewer 1 Report
A continuous gesture segmentation and recognition model is proposed in this manuscript. However, the model in this manuscript is a combination of the existing technology. This manuscript is more likely a technical report, not a paper. The novelty in this manuscript is not described in detail and paid more attention to. The detailed comments are as followed:
1. To reduce the model complexity, you choose to map the gesture motion in 3D complex space to 2D coordinate space. Why not directly use the RGB image instead of the projection? Moreover, Kinect could capture the keypoints of the human body, including the keypoints of the wrists. So, is it necessary for this manuscript to proceed with the target detection and the projection?
2. A comparison with existing hand gesture recognition technologies such as OpenPose and MediaPipe should be conducted in this manuscript.
3. The whole Section 4 is a detailed description of the existing technologies. Please present your improvements.
4. In Figure 10, the most important metric for the object/target detection algorithm is mAP. What is the meaning of “accuracy”? Yolov5 and CBAM are existing models. Have you made any improvements to these networks or modules? Please explain your improvements and present these comparative experimental data in the form of a table.
5. There is only one table in Section 5, please supplement the comparative experiments.

Author Response
See Word document for response.

Reviewer 2 Report
Please write the full name of G3-2D at the first occurrence of the term.
Please clearly describe the difference between continuous and dynamic gestures.
Section 2.1~2.2 requires additional reviews of the following other references.
>Girshick, Ross. "Fast r-cnn." Proceedings of the IEEE international conference on computer vision. 2015.
>Jeon, Haneul, et al. "Wearable Inertial Sensor-Based Hand-Guiding Gestures Recognition Method Robust to Significant Changes in the Body-Alignment of Subject." Mathematics 10.24 (2022): 4753.
To this reviewer's understanding, the research contributions of this paper seem to be two parts of gesture coordinate correction and time window-based gesture segmentation. However, as previously described in the reference papers below, motion detection and recognition based on sliding windows in the time-series data are widespread techniques.
> Jeon, Haneul, and Donghun Lee. "A New Data augmentation method for time series wearable sensor data using a learning mode switching-based DCGAN." IEEE Robotics and Automation Letters 6.4 (2021): 8671-8677.
Compressing the Spatio-temporal information of 3D space into a 2D plane is also a prevalent technique as in CNN, RNN, etc., and in this paper, the differentiation and progressivity compared to existing studies suitable for naming 'G3-2D' clearly can't deliver
Additional application studies should be added to assert the possibility of extending the method proposed in this study to other applications.
Author Response
See Word document for response.

Reviewer 3 Report
In the article, the authors presented their study called G3-2D: A Continuous Gesture Segmentation and Recognition Model to Shoot Gestures from 3DSpace to 2DCoordinateSystem”. The authors propose a G3-2D model to intelligently describe the recognition of continuous dynamic motions that map motions in 3D space to a 2D coordinate system using the target detection algorithm that preserves complex features in 3D space. Overall, it is a well-prepared article. My reviews and suggestions about their publications are listed;
The importance of the article and its contribution to the literature are not reflected in the abstract. The abstract should include the context or background information for your research; the general topic under study; the specific topic of your research; why is it important to address these questions; the significance or implications of your findings or arguments. It must also contain more numeric values. Please highlight your contribution. Reorganize the abstract to conclude:
(a) The overall purpose of the study and the research problems you investigated. (b) The basic design of the study. (c) Major findings or trends found as a result of the study. (d) A brief summary of your interpretations and conclusions.
Add more recent reference to enhance introduction section. Discuss the state-of-art techniques with their merits and issues. The literature should be developed. Discuss the research gaps and relate how the proposed work has improved them. There are many studies for pose estimation in the literature. For example, read this study; BabyPose: Real-Time Decoding of Baby's Non-verbal Communication using 2D Video‐based Pose Estimation.
What solution you propose to make the system more robust. What is your difference from similar studies?
YOLOv7, the newest YOLO algorithm surpasses all previous object detection models and YOLO versions in both speed and accuracy. It requires several times cheaper hardware than other neural networks and can be trained much faster on small datasets without any pre-trained weights. If possible, I suggest you try the YOLOv6 or YOLOv7 algorithms.
Some figure descriptions are really long. You should shorten them. For example; “Figure 6. Gesture trajectories of letter "a" under different algorithms: (a) is hand gesture trackgraphcollected without attention mechanism, (b) is hand gesture track map collected after usingtheattentionmechanism, and (c) is hand gesture trajectory graph collected after introducing gesture coordinatecorrection algorithm based on the attention mechanism”
Although some evaluation criteria are given in the article, It should be well supported by Precision, Recall (sensitivity), Accuracy, Specificity, Prevalence, Kappa, and F1-score. These results need to be analyzed, tabulated, presented graphically, and interpreted.
You must also provide numeric values. Rewrite the conclusion with following comments:(a) Highlight your analysis and reflect only the important points for the whole paper. (b) Mention the implication in the last of this section. Please, carefully review the manuscript to resolve these issues. (c) This section should be supported with numerical values.
Author Response
See Word document for response.

Round 2
Reviewer 2 Report
All of the revision requests suggested by this reviewer were well reflected in the revised manuscript.
Reviewer 3 Report
I have reviewed the revised manuscript title "Air-GR: An Over-the-air Handwritten Character Recognition System Based on Coordinate Correction YOLOv5 Algorithm and LGR-CNN". After revising my initial comments and comparing the changes, done by the authors, with them. I found that the authors addressed and answered most of the comments efficiently. Overall, the revised manuscrip is well organized and carefully prepared. The response letter was elegant and satisfactory. I thank the authors for their kind responses. The authors have sufficiently address my all comments. So, I think it is appropriate to accept the revised article. The authors have addressed all the concerns and responded to the review comments. The manuscript can be published in this journal.